# Characterising Rhythmic and Episodic Pulsing Behaviour in the Castleton Karst, Derbyshire (UK), Using High Resolution in-Cave Monitoring

John Gunn * and Chris Bradley 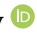

School of Geography, Earth and Environmental Sciences, University of Birmingham, Birmingham B15 2TT, UK; c.bradley@bham.ac.uk
* Correspondence: j.gunn.1@bham.ac.uk

**Abstract:** The discharge from most karst springs exhibits a consistent and reasonably predictable response to recharge but a few exhibit short-term ('rhythmic') changes in flow that are commonly attributed to the geometry of feeder conduits and the action of siphons. This paper investigates water flow in a karst system that exhibits rhythmic and episodic changes in discharge due to variations in flow from two phreatic conduits (Main Rising (MR) and Whirlpool Rising (WR)) that pass through Speedwell Cavern en route to the springs. Water tracing experiments indicate that the conduits receive both allogenic and autogenic recharge. Flow dynamics and conduit behaviour were investigated using high-resolution (2-min) water depth data collected from MR and WR between 2012 and 2015 (when MR was dominant) and between 2021 and 2023 (when WR was dominant). Water depths were also logged in a cave at the upstream end of a conduit draining to both MR and WR and at springs. The short-term temporal variability in water depths at both MR and WR is greater than any documented in previous studies. This is attributed to conduit bedrock geometry and changes in conduit permeability due to sediment accumulation in phreatic loops, which together influence the response to recharge.

**Keywords:** phreatic conduits; rhythmic karst springs; karst hydrogeology

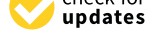



## 1. Introduction

The majority of karst groundwater systems are highly anisotropic and heterogeneous with three porosity/permeability elements: (1) intergranular; (2) fracture/fissure/bedding plane; and (3) conduit. Modellers commonly group the first two and refer to "the fissured rock matrix" as one porosity/permeability group with conduit porosity/permeability as a second group only found in karst. While groundwater is stored in the fissured rock matrix, the majority of water transmission occurs via the conduit network. Conduits only occupy a small proportion of the rock mass, so there is a low probability of intersection by boreholes, but they connect with the surface at springs. Consequently, springs are to karst hydrogeologists what boreholes are to hydrogeologists working in other lithologies, and many attempts have been made to develop models that quantify flow from springs as a function of effective precipitation over the catchment. However, spring discharge and system response to environmental change can only be modelled if spring output is a direct function of recharge. This is the case in most karst springs, but a small number of springs are characterised by relatively short-term (minute-by-minute to hourly) changes in flow (i.e., rise and fall) that are independent of recharge and are superimposed onto recharge-driven hydrographs. These have been variously referred to as ebbing and flowing springs, intermittent springs, periodic springs and rhythmic springs [1]. Rhythmic Karst Spring (RKS) is the most widely used term, although both periodic and rhythmic imply a degree of regularity that is not always present. This may be why Guo et al. [2] preferred the term Intermittent Karst Spring (IKS) for their model, which simulates flow from springs in which

the periodicity varies over time. However, the term intermittent spring is also commonly applied to karst springs that only flow for part of the year, including overflow springs, and hence there is potential for the two very different types of spring to be confused.

Irrespective of the terminology used, there is widespread agreement that springs that exhibit short-term changes in flow that are not directly related to recharge are globally rare and have been a subject of fascination for centuries. In one of the earliest descriptions in the scientific literature, Oliver [3] described an ebbing and flowing spring in Devon (UK), the Lay-Well (also Laywell) for which he subsequently timed the ebb and flow cycle [4]. Oliver's observations, and those of subsequent visitors to the site, are documented by Mather [5], who also notes that Atwell [6] proposed that the anomalous fluctuations could be explained by the operation of a siphon. This explanation has been adopted by subsequent researchers, most recently by Xiao and Zhang [7] and Guo et al. [2], both of whom built laboratory physical models to test their analytical models. Debieche et al. [8] carried out very short-interval (15 s to 5 min) visual observations of water depth and estimated velocity and discharge from a rhythmic spring in Algeria, but their observations were only over periods of a few hours. The ebb and flow behaviour of Big Spring in Kings Canyon National Park, CA, USA was investigated by Sara [9], Urzendowski [10] and Slattery and Hess [11], the latter reporting that digital stage data were collected 'almost continuously' over a 3-year period at an in-cave site and over a 6-year period at the spring. This is the only field study known to the authors that logged short-term variations in water depth at springs displaying anomalous flow over longer time periods.

While this paper seeks to address this gap, the research described herein began in 1985 as a doctoral research project on the hydrology of the Castleton karst (Derbyshire, UK). A broad-crested rectangular weir and Ott water depth recorder were installed on Peakshole Water (PW), which is fed by three karst springs, and the pen-and-ink trace revealed a series of anomalous changes in water depth. While the doctoral project was not completed, the water depth data were interpreted by Bottrell and Gunn [12], who suggested that the anomalous periods related to flow switching between two phreatic conduits in Speedwell Cavern, Main Rising (MR) and Whirlpool Rising (WR), that was caused by movement of an unstable sediment pile in the phreatic part of the system. At the time, no suitable instruments were available to monitor water depth within the karst system, and no further research on MR and WR was undertaken until 2012. On 25 February 2012, a party of cavers in the Speedwell Cavern streamway were surprised by a flood pulse that was completely unexpected, as their visit was during a period of settled dry weather [13]. This led to a 'citizen science' project to investigate why this had occurred. The cavers obtained charity funding to purchase data logging depth sensors, which were installed in MR and WR, and further funding from one of the authors of this paper (JG) allowed the monitoring network to be expanded to include other sites both underground and on the surface. The loggers were downloaded by cavers assisted by JG. This phase of data collection ran from July 2012 to April 2015, and throughout this period MR remained the dominant input.

In spring 2021, cavers reported very low flow from MR and that the major input had switched to WR. JG sought charity funding for a new monitoring network, and data collection commenced in June 2021. Data collected since then have confirmed the switch from MR to WR as the dominant input with consequent changes to hydrograph characteristics. In the present paper, our aim is to (i) summarise data collected during both periods (i.e., 2012–2015 and 2019–2022); (ii) to investigate the phreatic conduits and (iii) to demonstrate the potential of in-cave monitoring data to explain the behaviour of karst springs, as discussed by Skoglund et al. [14] in this Special Issue.

## 2. Materials and Methods

### 2.1. The Castleton Karst

Castleton (Derbyshire) is situated on the northern margin of the Peak District karst (Figure 1). The bedrock geology is dominated by a thick succession of limestone beds of Viséan (early- to mid-Carboniferous) age that belong to the highly karstified [15,16] Peak

Limestone Group, which is part of the Carboniferous Limestone Supergroup [17]. To the north, the limestones dip steeply beneath mudstones of the Bowland Shale Formation which are overlain by Quaternary solifluction deposits. Millstone Grit Group rocks crop out further north on Rushup Edge (Figure 2). A series of streams with a combined catchment of c. 5 km$^2$ flow over the solifluction deposits before sinking into the marginal limestones at 15 discrete points (Figure 2). There is no surface drainage in the area underlain by limestones, and hence precipitation in this area contributes to dispersed autogenic recharge.

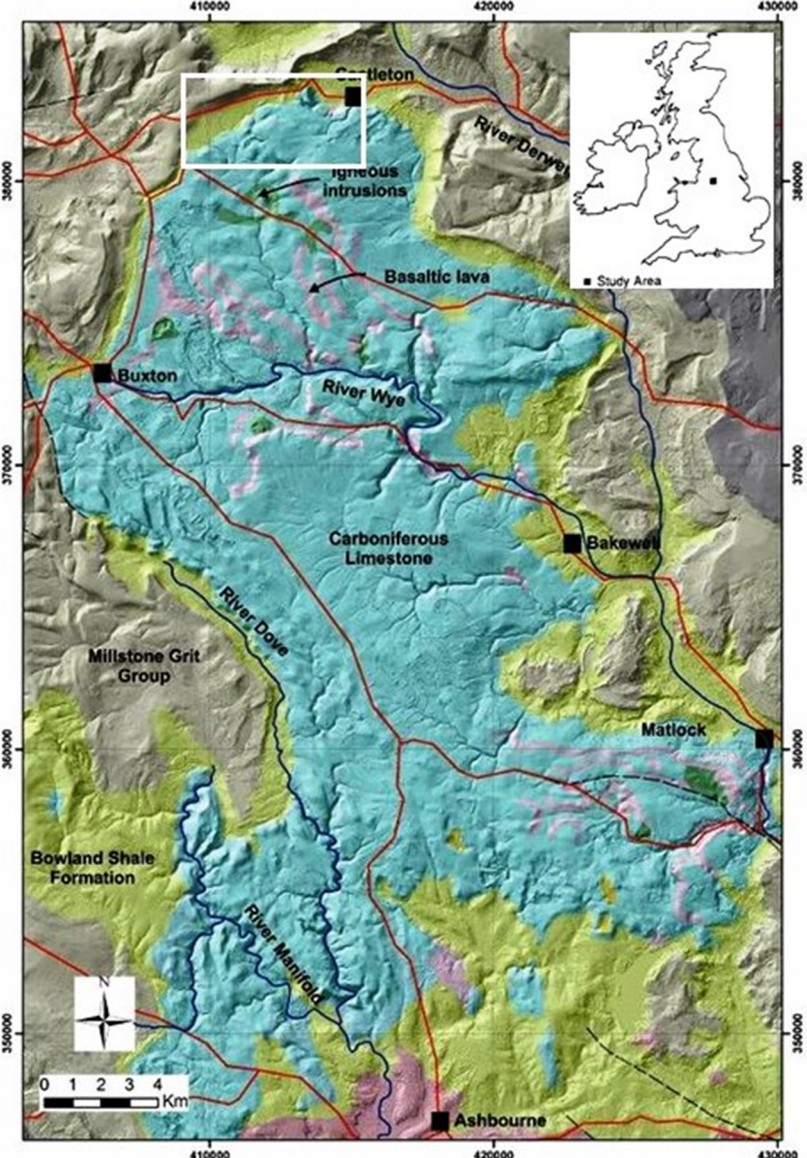

**Figure 1.** Geology of the northern Peak District karst region, with field site location indicated by the rectangle (in white) and the location in the UK shown in the inset. (Based upon 1:625,000 scale DigMap bedrock geological maps, with the permission of the British Geological Survey. NEXTMapTM Britain elevation data from Intermap Technologies).

The Castleton karst contains the most extensive and complex karst drainage system in the Peak District and is a designated Site of Special Scientific Interest [18]. There are no anthropogenic activities in the catchment that involve discharge of water to or abstraction of water from the karst aquifer. Within the system there are 45 known caves with a total passage length of c. 30 km, the longest being the Giant's–Oxlow cave system and the Peak–Speedwell cave system (Figure 3). Underground flow directions can be inferred from the results of >50 water tracing experiments completed in the Castleton karst, e.g., [19,20]

which are summarised in Figure 2. There is no relationship between surface topography and underground flow which passes beneath surface drainage divides (Figure 3) [15,19,20]. Tracer injected into sinking streams (P0–P12) and into the autogenic percolation-fed streams in Eldon Hole, Nettle Pot, Winnats Head Cave, Blue John Cavern and Treak Cliff Cavern has been found to emerge from Russet Well (RW) and Slop Moll (SM), two springs on opposite sides of the Peakshole Water (PW). This river (PW) has its source at Peak Cavern Rising (PCR) which discharges autogenic recharge for much of the year but acts as an overflow spring at times of high flow. In most of the water tracing experiments, only the springs were monitored, but in those cases where there was underground monitoring, tracer injected into the allogenic stream-sinks and into the autogenic streams sinking in Nettle Pot and Winnats Head Cave was detected at one or both of two sumps (water-filled conduits) in Speedwell Cavern: Main Rising (MR) and Whirlpool Rising (WR). As the dye was detected in both sumps [19], the conduits draining from the sinking streams must first unite and then bifurcate, as shown in Figure 2.

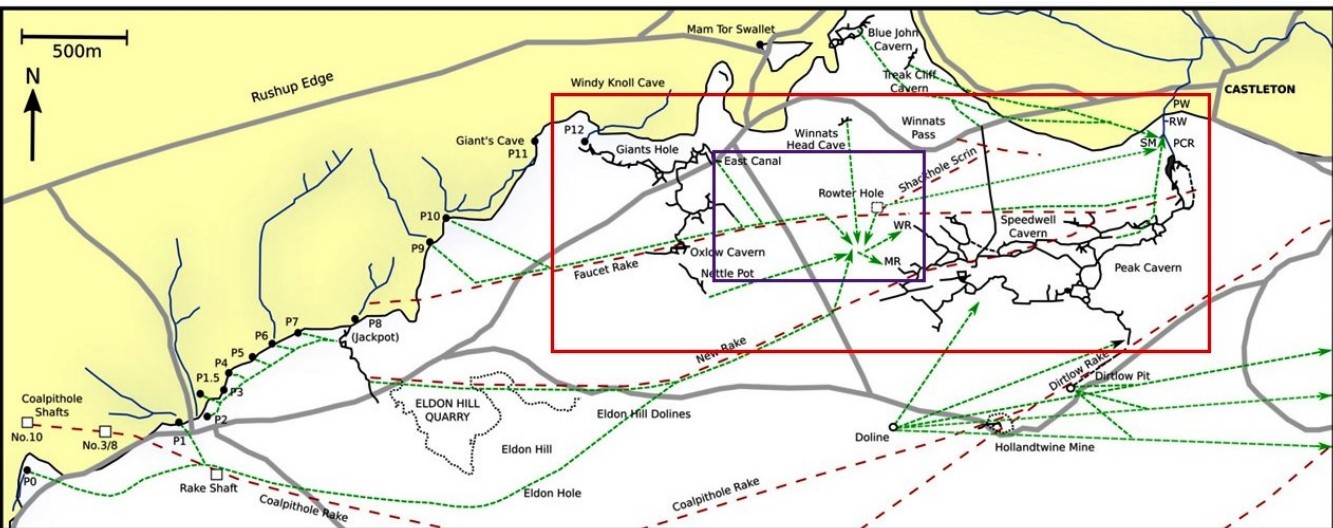

**Figure 2.** The Castleton karst system showing surface drainage on the Millstone Grit (yellow shading) to the north, and the sub-surface conduit network in the Carboniferous Limestone. Black lines are surveyed cave passages, and green lines show hypothetical pathways of links proven by dye tracing. The red lines show fault-aligned mineral rakes. The conduit network in the red rectangle is illustrated in Figure 3, and the area in the purple rectangle is shown in Figure 4.

Exploration by cave divers has revealed that the MR conduit has a complex bedrock profile (Figure 4) and at its furthest explored point, 74 m below the rising, water is described as 'boiling-up' through a floor of liquid sand from a slot around 2 m wide [21]. The water elevations in MR and WR display complex pulsing behaviour (described below) but in general the base elevation at MR is c. 232 m above sea level (asl) and WR is c. 11.5 m higher. WR has also been explored and surveyed by cave divers who found that initially it has a more horizontal profile than MR. However, water enters WR from a vertical shaft that probably descends to greater depth but is constricted and has yet to be passed by divers (Figure 4). Approximately 1050 m NW of MR, the water sinking at P12 enters East Canal, the downstream sump in Giant's Hole Cave. The lowest observed water elevation at the sump is 241 m asl, and the water elevation has been observed to increase by 23 m above this (Figure 4). The lowest explored point in the sump is at c. 221 m asl, where the floor is described as soft silt with some boulders [22].

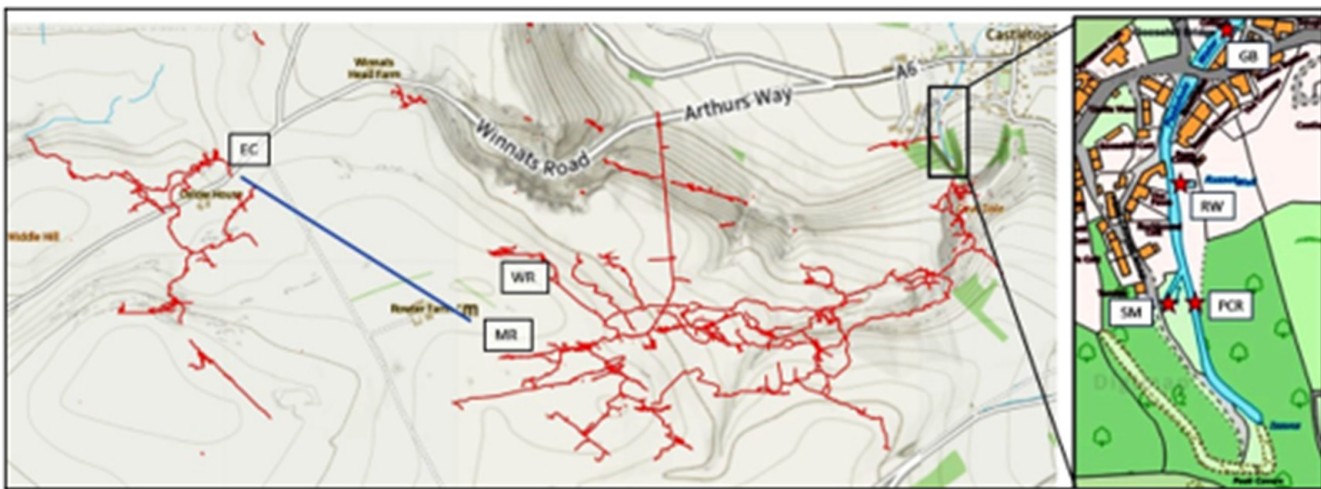

**Figure 3.** The Giant's Hole (**left**) and Peak–Speedwell cave systems and location of underground and surface monitoring sites. EC = East Canal; MR = Main Rising; WR = Whirlpool Rising; PCR = Peak Cavern Resurgence; SM = Slop Moll; RW = Russet Well; GB = Goosehill Bridge. Blue line shows location of cross-section in Figure 4. Main figure is from https://peakdistrictcaving.info/home/the-caves/castleton/map, accessed on 15 April 2023. Inset is from Digimap (© Crown copyright and database rights 2023 Ordnance Survey (100025252)).

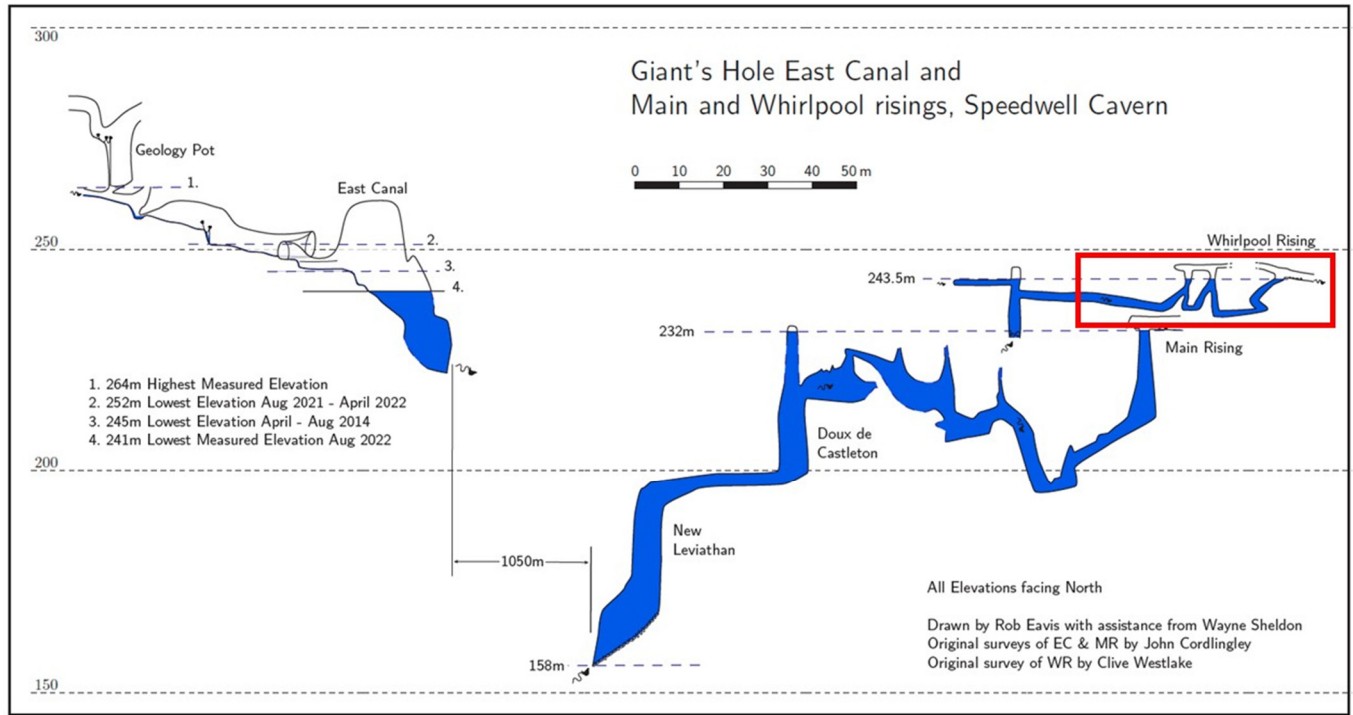

**Figure 4.** Surveyed profiles of Giant's Hole East Canal (EC) and the Main and Whirlpool risings (MR and WR). Elevation is in metres above sea level, and WR is 225 m north of MR (see Figure 3). Inset (in red) is discussed in Section 4.

Downstream of MR, the water flows along open vadose stream passages for c. 150 m before passing through collapse debris (the Boulder Piles) which effectively act as a permeability barrier that influences upstream water depth. Downstream of the Boulder Piles, there is a further 150 m of open vadose passage leading to a 1.5 m to 2 m deep lake, 'The Whirlpool'. The water from WR flows for 300 m along a narrow meandering vadose passage before joining the stream from MR at The Whirlpool. The combined flow then flows

along an open passage for up to 930 m before entering the Downstream Sump (DS), which has been explored by divers for a further 150 m. DS has a limited capacity, and during periods of high flow water backs up, eventually completely filling the cave passage for up to 400 m upstream of the sump and rising into overlying passages in Peak Cavern. From the limited exploration that has been possible to date, there is at least 400 m of water-filled conduit to Slop Moll and a further 60 m to Russet Well.

### 2.2. Data Collection

In situ data collection commenced in June 2012 when cavers installed pressure sensor loggers (10 m range, +/1 cm accuracy) in plastic pipe stilling wells at MR and WR. The loggers were initially programmed to sample at 1 min intervals, as the intention was to investigate short-term changes in water depth. At this monitoring frequency, the logging capacity was reached after 31 days, and there are occasional gaps in the data series when it was not possible to visit MR and WR to download data before the logger storage capacity was reached. To address this, in March 2014 the logging interval was increased to 2 min resolution and sampling continued until April 2015. As the loggers record total pressure, an atmospheric pressure sensor was installed at the Bottomless Pit in Speedwell Cavern so that water depths at MR and WR could be obtained by subtracting atmospheric pressure from total pressure. The datum for measurements at each site were one metre rulers fixed to Dexion steel strips and bolted to the passage wall (Figure 5). These were later tied into a cave survey, allowing water depths to be expressed as metres above sea level (asl). A second phase of 2-min data collection at MR and WR commenced on 26 July 2021 and is ongoing.

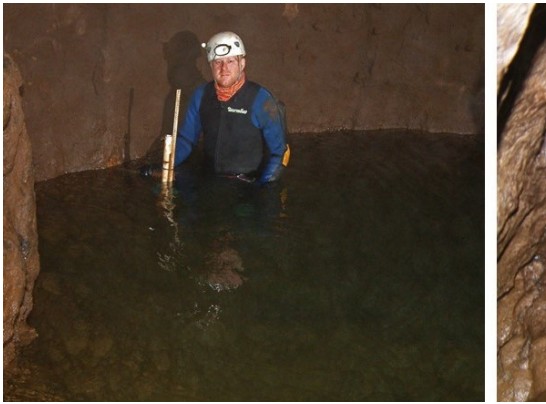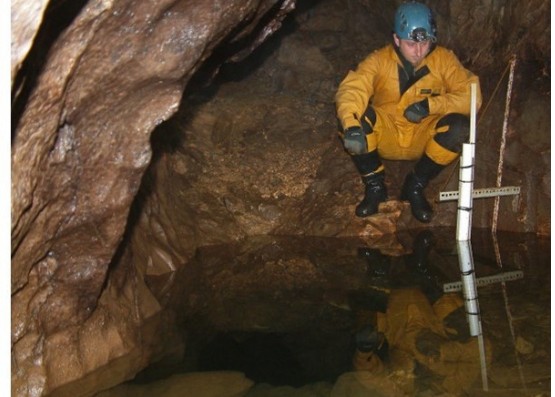

**Figure 5.** Stilling wells installed at Main Rising (**left**) and Whirlpool Rising (**right**).

Additional monitoring was undertaken in a conduit that drains to both MR and WR and which has its upstream end at the East Canal (EC) sump in Giant's Hole (Figures 3 and 4). A pressure sensor with a 20 m range (+/− 2 cm accuracy) and logging at 2-min intervals was installed in the sump by cavers on 8 April 2014 and removed on 13 August 2014. The logger location was tied into a cave survey enabling water elevation (m, asl) to be calculated. A second phase of data collection at EC commenced on 20 August 2021 and is ongoing, with the logging interval increased to 4 min to maximise the time that the sensor can be left underground before reaching the logger storage capacity (c. 4 months). For comparison with other sites where depth was logged at 2-min intervals, a 2-min data set was constructed by direct interpolation between the 4-min data.

On the surface, a combination of pressure sensors and capacitance loggers was used to monitor water depth at three springs: Peak Cavern Rising (PCR), Slop Moll Rising (SM) and Russet Well (RW), and at a weir on the Peakshole Water (PW) downstream of the springs. From September 2013 to August 2014, a tipping-bucket rain gauge with logger was maintained in Sparrowpit on the catchment boundary, and from August 2020 a tipping-bucket rain gauge with logger was located on the top of Coalpithole Shaft No. 10 (western

edge of Figure 2), which is 600 m north of Sparrowpit. The Environment Agency has also maintained a telemetric tipping-bucket rain gauge at Chapel en le Frith, 2300 m SW of Sparrowpit, since 2006, and data from this gauge were used to fill in occasional gaps in the data series.

At all sites, the loggers were synchronised with the clock on a laptop computer at the time of each download. The clock was regularly checked and was maintained at Coordinated Universal Time (UTC).

## 3. Results

During the two phases of data collection at Castleton, several hundred thousand data points were recorded at seven sites. Here we present a selection of data from two primary data series to illustrate a variety of system responses to recharge: (1) 18 December 2013–16 June 2014, when MR was the dominant input to Speedwell Cavern, and (2) 21 August–6 November 2021, when WR was the dominant input. Each data series is first graphed as a whole before examining shorter time periods in detail. In all data sets, there are occasional changes in water depth that appear not to reflect recent recharge, and these are described using the following terms. **Rhythmic** changes are those in which the depth changes are wave-like with a clear **peak** (greatest depth) and a clear **trough** (lowest depth). In contrast, **episodic** changes are those in which the depth falls (or rises) rapidly and then remains at broadly the same level for a period of time at the end of which there is another rapid rise (or fall). Where the level is low relative to the overall trend, the form resembles an **elongated trough,** and where the level is high relative to the overall trend the form resembles a **plateau**. The time between the end of the fall (or rise) and the start of the rise (or fall) is the **duration** of the elongated trough (or plateau). When depth changes are rhythmic, the waveform is commonly uneven in that the time between successive peaks differs from the time between successive troughs. This makes it difficult to identify a wavelength ($\lambda$) or time period (T) in the sense in which these terms are commonly used. As there are also variations in the number of peak–trough cycles in a given time period, we use the term **frequency** in a broad sense, whereby high frequency is indicative of a greater number of peak–trough cycles per unit time. The peak–trough cycles are commonly superimposed on a rising or falling trend, and in the absence of a clear 'rest level' the **amplitude** is defined as the difference in depth between successive peaks and troughs rather than the difference between rest position and peak.

### 3.1. Primary Series 1: 18 December 2013–16 June 2014

Hourly rainfall for 18 December 2013 to 16 June 2014 is presented in Figure 6A, with Figure 6B,C showing 2-min resolution water depths at PCR and RW (6B) and at MR and WR (6C). The depth hydrographs show a 'spikey' response that is typical of karst groundwaters that receive rapid recharge. However, the data are 'fuzzy' and there are five anomalous episodes, marked by * on the MR plot in Figure 6C, during which the depth appears as a vertical line that does not correspond to rainfall. These events are also seen in the RW data (Figure 6B). Two sub-sets of data have been identified to illustrate the complexity of the response: 18 December–20 December 2013 and 26–28 February 2014. From 8 April 2014 to 16 June, additional data are available from the East Canal in Giants Hole, and these are discussed as a third sub-set.

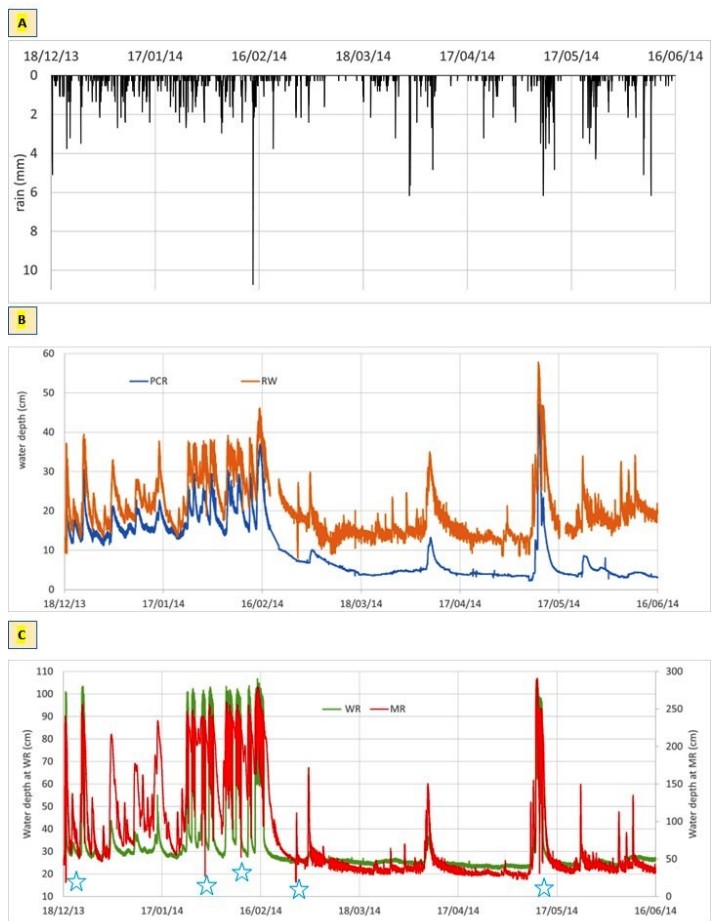

**Figure 6.** (**A**) Precipitation; (**B**) water depths at Peak Cavern Rising (PCR) and Russet Well (RW) and (**C**) water depth at Whirlpool Rising (WR) and Main Rising (MR), December 2013 to June 2014.

3.1.1. Subset 1: 18–20 December 2013

Figure 7A shows data for the 48-h period commencing 14:00 18 December. Initially, there was rhythmic periodicity at MR that was mirrored at SM and, with lower amplitude, at RW, but no rhythmic fluctuations in water depth were evident at WR. Between 18:00 and 22:00 on 18 December, a total of 11 mm of rainfall fell in the catchment during a short intense rain event. This was followed by a period that was largely dry except for three small events which together yielded only 2.1 mm of precipitation. Water levels began to rise at MR at 20:00 and peaked at 01:08 on 19 December, by which time water depths had increased by 192 cm (0.62 cm/min). At WR, water depth also peaked at 01:08, but the depth started to increase 90 min later than at MR, and the increase in depth was only 16 cm (0.07 cm/min). At the Castleton springs directly connected to the Speedwell conduit (RW and SM), water depths started to increase at 20:24 and peaked c. 01:50 on 19 December.

Following the peak, water levels at MR decreased slowly (mean 0.076 cm/min) for 156 min before dropping rapidly (mean 5.5 cm/min) for 38 min. There was then an elongated trough during which the water depths at MR remained constantly low for 176 min. This ended with a 20-min period of rapidly increasing water depths (mean 8.4 cm/min). The rapid drop in water depth at MR was mirrored at SM, where the fall began 12 min after MR and the mean rate was 0.99 cm/min, and at RW, where the fall began 12 min after MR and the mean rate was 0.48 cm/min. At RW, there was an elongated trough which ended 18 min after the MR trough, and the mean rate of increase was 0.71 cm/min. At WR, water depths were initially unchanged, but 6 min after the start of the elongated trough at MR they began to increase rapidly and after a further 6 min had risen 57.3 cm at a mean rate of 9.55 cm/min. Water depths at MR and WR for 8 h from 04:00 are shown in more detail in Figure 7C. After the sharp rise at WR, the water depth began a rhythmic

cycle. Over this hour, the time from peak to peak varied from 6 min to 8 min and the amplitude steadily decreased.

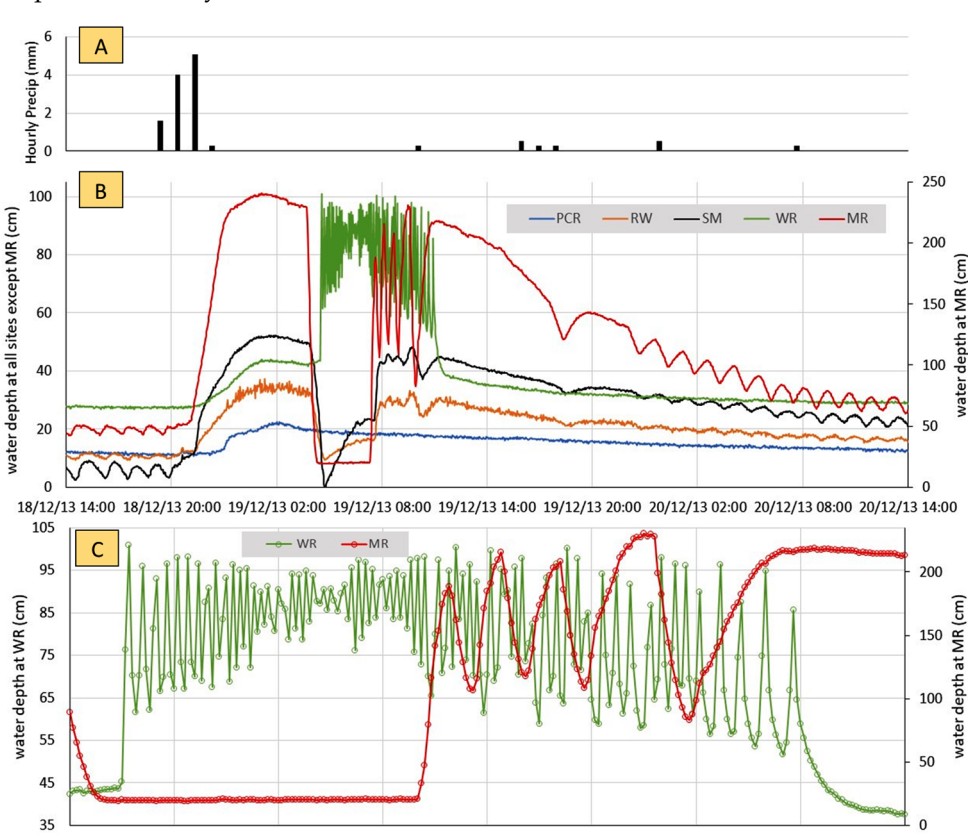

**Figure 7.** (**A**) Hourly precipitation (mm) and (**B**) two-minute water depths at PCR, RW, SM, WR, and MR for the 72-h from 14:00 on 18 December 2013. (**C**) Two-minute water depths at MR and WR for 8 h from 04:00 on 19 December 2013 (there was 0.3 mm precipitation between 09:00 and 10:00).

Following the post-elongated trough maximum depth at MR, there were four complete peak–trough cycles, each with a different time interval between peaks and a different amplitude. The cycles were also present at RW and SM but with much lower amplitudes, although there was a marked trough at both springs 24 min after the final MR trough. At MR, this trough was followed by 78 min of depth increase to broadly the same level as in the previous cycles. There was then a slow reduction in depth that followed a concave pattern with no rhythmic changes for 384 min (mean rate 0.18 cm/min), after which there was a more rapid decline (0.62 cm/min) to a marked trough. Water levels then increased for 104 min (average rate 0.21 cm/min) before peaking and then declining. After a further 120 min, a new rhythmic cycle started, which was superimposed on the recession curve. No cycle was apparent at RW or WR, where depths continued to fall very slowly. It Is clear from Figure 7A,B that during the period when there were rhythmic changes at MR and WR, the WR cycles have a shorter frequency than at MR, and both cycle frequency and amplitude change over time. Despite this apparent independence, the rhythmic changes at WR ceased at the same time as at MR. RW (which is directly connected to the Speedwell Cavern downstream sump that is fed by MR and WR) exhibited depth patterns that are distinct from PCR, where water depths increased gradually after the rain event but with a slower recession. This behaviour is typical of a spring fed by autogenic recharge.

### 3.1.2. Subset 2: 26–28 February 2014

During the 48-h period shown in Figure 7A, the changes in water depth form part of a single storm hydrograph, but episodic changes and periods with rhythmic changes

also occur during base flow recession, although these are difficult to characterise given the scale in Figure 6B,C. One such event is shown in Figure 8, in which changes in water depth are plotted over 48-h from 12:00 on 26 February 2014. Over the previous 7 days, there were occasional light showers with 10.7 mm of precipitation, and over the 48 h plotted in Figure 8 a further 7.8 fell, of which 5.3 mm was between 02:00 and 05:00 on 27 February. For the first 14.5 h, water depths at all sites fell very slowly as would be expected in a period of base flow recession. However, from 02:28 27 February, water depths at MR fell rapidly (0.55 cm/min) for 50 min and then remained broadly constant as an elongated trough for 208 min. Water depths then started to increase rapidly (0.57 cm/min), rising by 90.4 cm in 160 min and peaking at levels that were 63.7 cm higher than the initial base level. At RW, water depths started to fall 40 min after MR and continued to fall for 62 min at a mean rate of 0.1 cm/min. The elongated trough at RW lasted for 234 min, and water depths started to increase 58 min after the increase at MR, peaking after 110 min (0.17 cm/min). As was the case at MR, water depths at RW at the end of the rise were markedly higher than the base level before the start of the fall. After the peak, water depths at MR fell for 118 min at a mean rate of 0.39 cm/min before entering a period of rhythmic changes superimposed on an overall recession. Trends at RW were similar, with water depths falling for 140 min (mean rate of 0.04 cm/min) before a rhythmic cycle commenced. The period of rhythmic changes at MR and RW continued for 14 h, after which the depth trends were similar to those observed between 12:00 26 February and 02:28 27 February at the start of the plotted data series.

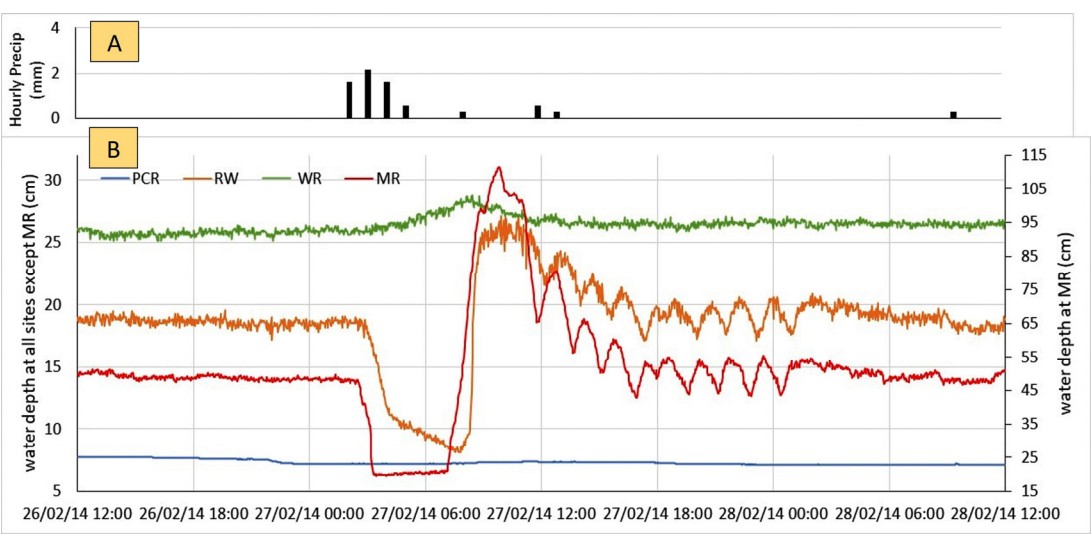

**Figure 8.** (**A**) Hourly precipitation (mm) and (**B**) 2-min water depths at Main Rising (MR), Peak Cavern Rising (PCR), Russet Well (RM) and Whirlpool Rising (WR) for 48 h from 12:00 on 26 February 2014.

At WR, water depths declined slowly from 12:00 26 February to 02:28 27 February, but when depths at MR started to fall rapidly, water depths at WR then started to rise slowly (0.008 cm/min) through to 08:26 27 February before subsequently falling equally slowly. However, by 12:00 on 28 February water depths at WR were still 0.6 cm higher than at 02:28 on 27 February. At PCR, the range in water depths between 12:00 26 February and 12:00 28 February was only 0.7 cm, indicating the absence of any recharge during this period of episodic and rhythmic changes at MR.

### 3.1.3. Subset 3: 8 April–16 June 2014

The conduits that discharge at MR and WR extend upstream to the Rushup Edge stream-sinks (Figure 2), and the upstream end of the conduit fed by the P12 sink is accessible as the East Canal (EC), the downstream sump in Giant's Hole (Figure 3). Figure 9B shows 2-min resolution water depths in EC, MR, WR, RW and PCR from 8 April to 16 June 2014

with hourly rainfall plotted at the same scale above (Figure 9A). Between 8 April and 8 May, water depths at all sites fell very gradually with little or no change following the rainfall events. However, at EC, MR and (less obviously) RW, rhythmic changes were superimposed on this trend. Between 12:00 on 8 May to 00:00 on 13 May, 77 mm of rain led to substantial increases in water depth (discussed below). Following this event there was a change in rhythmic frequency that is most evident at EC but with an amplitude that remained similar to the pre-storm period.

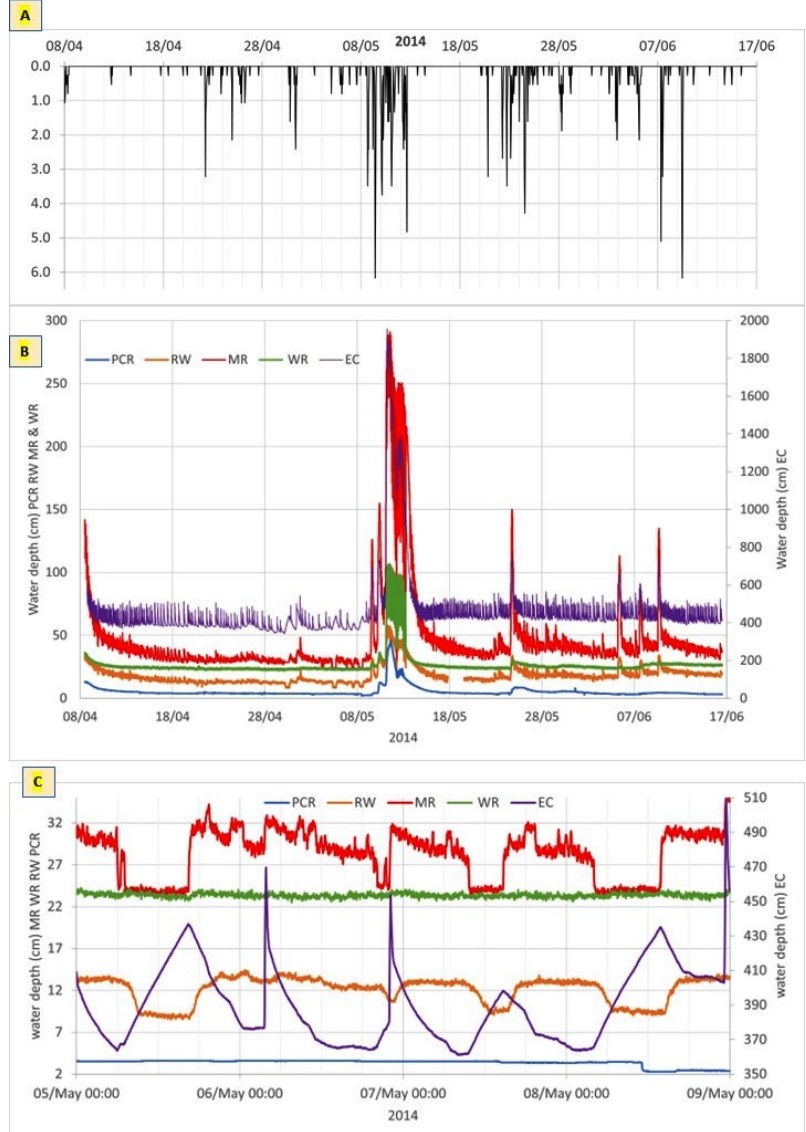

**Figure 9.** (**A**) Hourly precipitation (mm) and (**B**), 2-min water depths at East Canal (EC), Main Rising (MR), Peak Cavern Rising (PCR), Russet Well (RM) and Whirlpool Rising (WR) from 8 April to 17 June 2014. (**C**) 2-min water depths at EC, MR, PCR and WR from 5–8 May 2014.

In addition to the rhythmic behaviour, there were six elongated troughs similar to those shown in Figures 7 and 8 but with a markedly smaller fall in depth. Five of these troughs, which occurred between 5 and 8 May, are shown in Figure 9C with descriptive statistics tabulated in Table 1. The troughs were of different durations, and whilst the trough base level during events 1, 3, 4 and 5 was approximately 24 cm, event 2 had a trough base level that was 5 cm higher. One common factor is that in each case the start of the fall was within 4 min of water depths at EC starting to rise above their minimum level. Similarly, the water depth increases at the end of each elongated trough began within

4 min of water depths at EC starting to fall after attaining their peak. Two types of water level rise and fall at EC are apparent in Figure 9C. During events 1, 4 and 5, there was a steady linear rise followed by a steady linear fall, but during events 2 and 3 the rise is very rapid (4.65 cm/min) before initially falling rapidly and then at a more gradual rate. Each elongated trough at MR is also evident at RW but with a varying lag (Table 1) and dimensions (Figure 9C). Water depths at WR and at PCR remained virtually constant throughout the monitoring period.

**Table 1.** Information on five elongated trough episodes in the period 5–8 May 2014.

| Event | Start | End | Start Depth (cm) | Depth in Trough (cm) | Trough Duration (hh:mm) | Rate of Fall (cm/min) | Rate of Rise (cm/min) |
|---|---|---|---|---|---|---|---|
| 1 | 05/05/14 05:54 | 05/05/2014 16:54 | 31.50 | 24.00 | 10:20 | 0.26 | 0.26 |
| 2 | 06/05/14 00:28 | 06/05/2014 03:48 | 31.90 | 29.00 | 02:56 | 0.23 | 0.31 |
| 3 | 06/05/14 18:46 | 06/05/2014 22:14 | 30.50 | 24.40 | 01:46 | 0.50 | 0.50 |
| 4 | 07/05/14 09:36 | 07/05/2014 14:58 | 27.80 | 24.30 | 05:00 | 0.29 | 0.29 |
| 5 | 08/05/14 03:58 | 08/05/2014 14:04 | 27.60 | 23.70 | 09:30 | 0.38 | 0.38 |

Water depths at EC, MR and RW during the major storm event that followed the period of low flow shown in Figure 9C are plotted in Figure 10. Between 00:00 on 9 May and 12:00 on 10 May, the depth at WR and at PCR was virtually constant, and the depth changes at EC, MR and RW were smooth and linear with no short-term rhythmic changes.

During this period, the depth at EC increased by >2 m in two events, which are also seen at MR and RW and appear to be in response to rainfall, although there was no change in depth at WR and PCR. In the first event, the depth at EC started to rise at 10:16 on 9 May; at MR, the rise began at 10:50, and at RW the rise began at 11:32. In the second event, the delay between the two sites was markedly less, as the depth rise began at 02:28 10 October (EC), 02:30 (MR) and 03:00 (RW) with peaks at 09:10 (EC) and 10:04 (MR). Unlike in other events, the depth at RW did not rise to a clear peak but reached a plateau between 09:52 and 10:30. At all three sites, there is a steady recession which ends at 13:12 (EC), 13:16 (MR) and 13:30 (RW). Between these times and 11 May at 01:20 (EC), 01:32 (MR) and 01:46 (RW), there were six smaller rhythmic peaks and troughs.

After 01:20 11 May, there was a rapid increase in depth at all sites, most notably at EC, where the depth increased from 555 cm to 1956 cm at a mean rate of 5.5 cm/min. The rise was initially smooth but rhythmic peak–trough cycles began at 03:26 (WR) and 03:56 (MR and EC). The pattern is complex, but in broad terms the troughs at MR corresponded to peaks at EC and at WR. At WR, the peak–trough cycles had essentially the same amplitude until 04:30 on 12 May, but as the depth decreased at MR the amplitude of the peak–trough cycles increased both here and at MR. The pattern changed at 04:04 on 12 May, as after a peak at EC (with a corresponding trough at MR) there was a slow fall in depth until 05:48. During this period of falling depth at EC, the depth at MR increased and then stabilised as a plateau. At 05:48, the depth began to increase at EC and reached a peak at 06:34. At MR, the plateau ended, and water depth began to fall 6 min before the depth began to increase at EC, the first instance where a major change at MR preceded an equivalent change at EC. Following the trough at MR, the rise–plateau–fall cycle was repeated twice, and on both occasions there was a 'fall–elongated trough–rise' cycle at EC. After these episodes, the depth at both EC and MR began a rhythmic cycle which ended with two marked troughs at MR that were separated by a plateau and which coincided with peaks in the depth at EC. The final trough at MR was followed by a gradual increase in depth for 80 min and then a steady decline with no rhythmic change for c. 9.5 h. The final peak at EC was also followed by a steady decline which continued for 10 h 50 min. A new period of rhythmic changes began at the end of the steady fall, and in contrast to the previous periods of rhythmic change where the MR peaks were broadly coincident with EC troughs, the two sites follow a similar pattern but with the EC peaks and troughs 4–8 min before the corresponding peak and trough at MR.

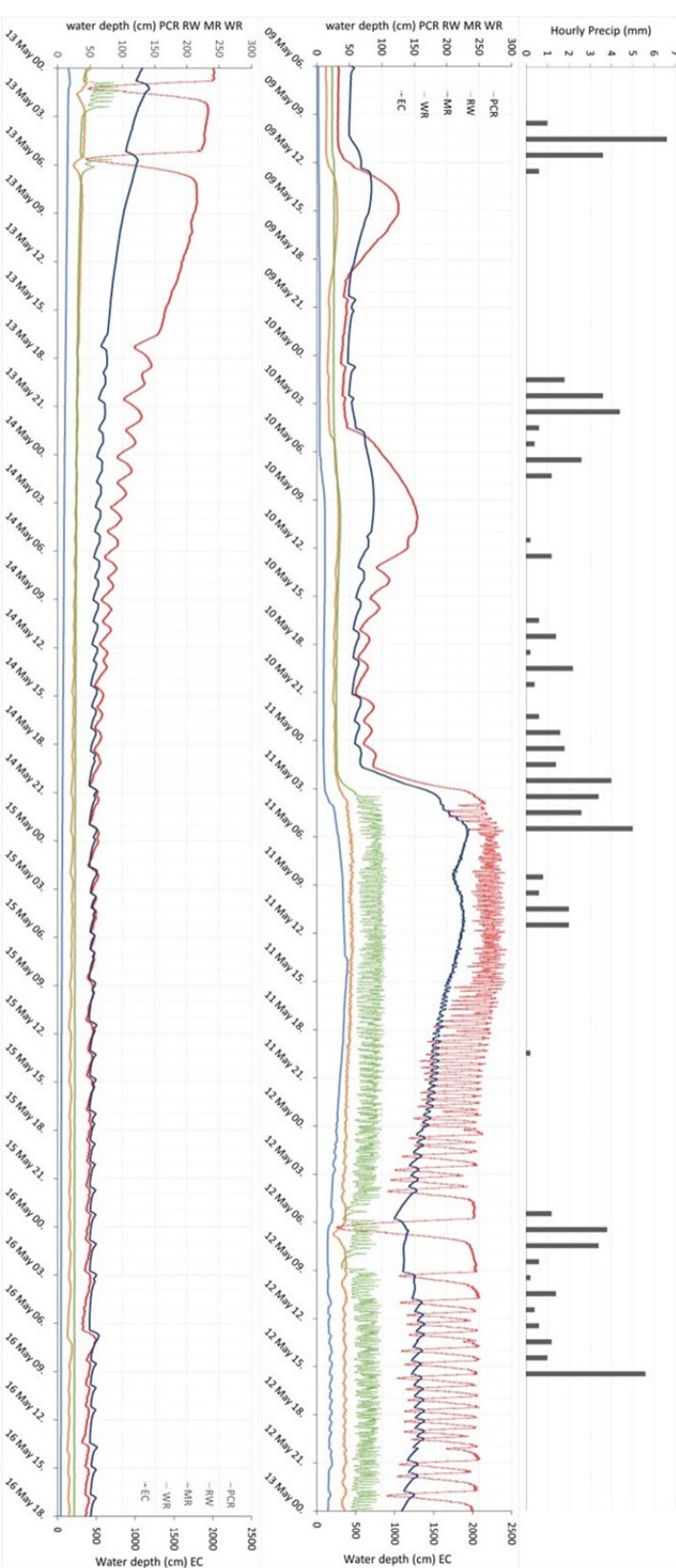

**Figure 10.** Hourly precipitation (mm) and 2-min resolution water depth at East Canal (EC), Main Rising (MR), Peak Cavern Rising (PCR) and Whirlpool Rising (WR) from 06:00 9 May to 00:00 13 May (**left**) and water depth at the same sites from 13 to 16 May 2014 (**right**). (There was no precipitation from 13 to 16 May).

At WR, the rapid increase after 01:20 11 May was followed by rhythmic periodicity, which continued with broadly the same frequency and amplitude until 02:24 on 13 May and completely ceased at 06:08, after which the depth fell very slowly with no rhythmic changes. During the period of rhythmic change, the frequency at WR was initially broadly similar to that at MR, but at MR there was a decrease in frequency and an increase in amplitude whereas at WR they remained broadly the same. However, there was a marked reduction in frequency and a smaller reduction in amplitude at WR during the plateau episodes at MR, and the final peak at WR coincided with the final deep trough at MR and peak at EC. There were no peak–trough cycles at WR during the final phase of the recession, when the peak–trough cycles at EC and MR were broadly synchronous.

### 3.2. Primary Series 2: 21 August–6 November 2021

Hourly rainfall for 21 August to 6 November 2021 is presented in Figure 11A, and 2-min resolution water depths at EC, MR and WR are plotted in Figure 11B. At all three sites, the depth hydrographs differ significantly from those shown previously in Figures 7 and 8. At EC, the minimum observed depth is over 1 m below the minimum in 2014, and at MR the maximum depth over the 2021 study period (15.9 cm) is less than the minimum depth between 18 December 2013 and 16 June 2014 (19.2 cm). For the majority of the time there is very little change in water depths at MR. In contrast, WR displays rhythmic behaviour with variable frequency and amplitude for most of the study period, although there are intermittent periods when there is very little change in depth. There is a very close relationship between water depths at EC and WR, although the nature of the relationship varies with water depth. In the following section, two subsets of data are summarised to illustrate the variable relationship with depths, 30 August–5 September and 27 September–3 October.

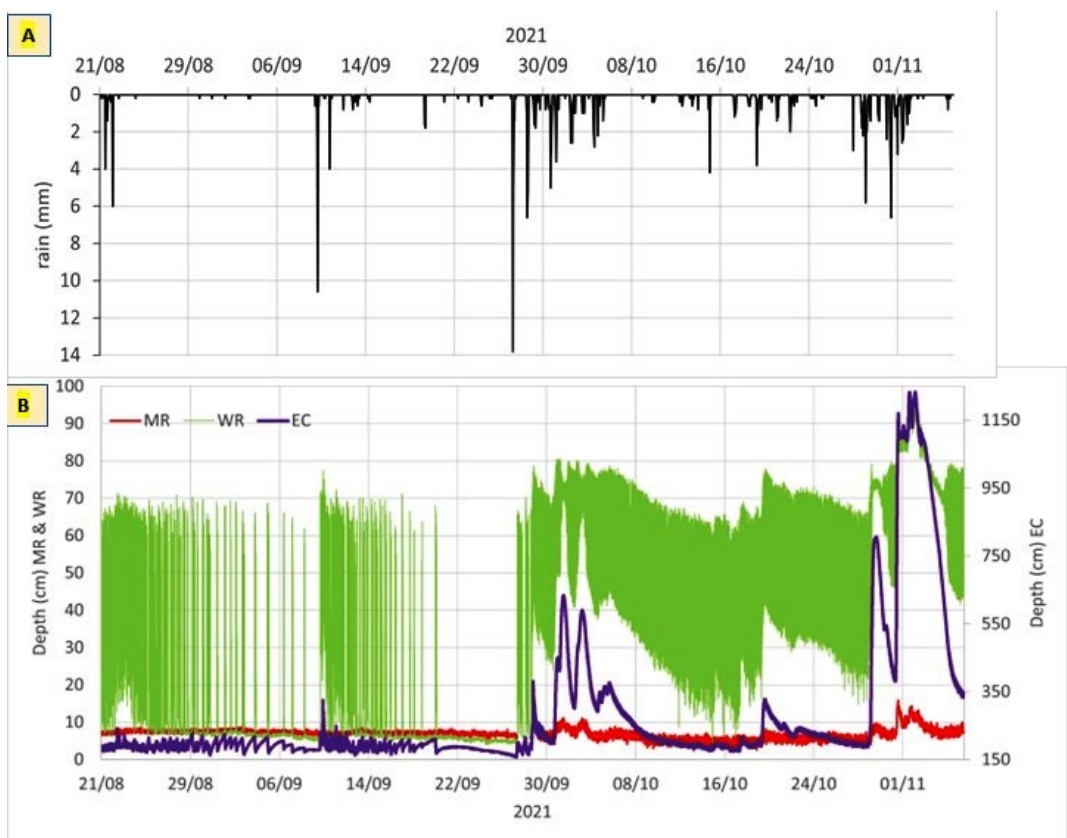

**Figure 11.** (**A**) Hourly rainfall and (**B**) 2-min water depths at East Canal (EC), Main Rising (MR) and Whirlpool Rising (WR) (**B**) from 21 August to 6 November 2021.

### 3.2.1. Subset 1: 30 August–5 September 2021

Water depths from 30 August to 5 September 2021 are shown in Figure 12A. Over this period variations in water depth at MR were less than +/− 1 cm, the accuracy of the logger. At EC there were 10 peak–trough cycles, details of which are provided in Table 2. In all cases, the rate of fall was more rapid than the rise, but there was no consistent pattern. The frequency of peaks (time between successive peaks) ranged from 3 h 58 min to 27 h 48 min, and the trough frequency ranged from 6 h 00 min to 28 h 16 min; the rates of rise and fall, respectively, ranged from 0.03–0.09 and from 0.14–0.58 cm/min. Within 2 min of each peak at EC, the water depth at WR began to rise and enter a rhythmic cycle that continued until the next EC trough. Within 2 min of each EC trough, the water depth at WR began to fall, and this ended when the depth reached a base level or declined very slowly until the next EC peak. In eight of the nine WR base level periods, the duration was 26–36 min, less than the EC rise time (Table 2), but during event 5 the rate of fall at WR was much slower than in the other events, and consequently, the base level duration was 122 min less than the EC rise time. The frequency and amplitude of the rhythmic cycles at WR, which occurred while the depth was falling at EC, varied between events, but detailed analysis (e.g., Figure 12B) shows that small peaks and troughs were superimposed on the overall EC recession and that each minor EC peak coincided with a WR trough and each minor EC trough coincided with a WR peak.

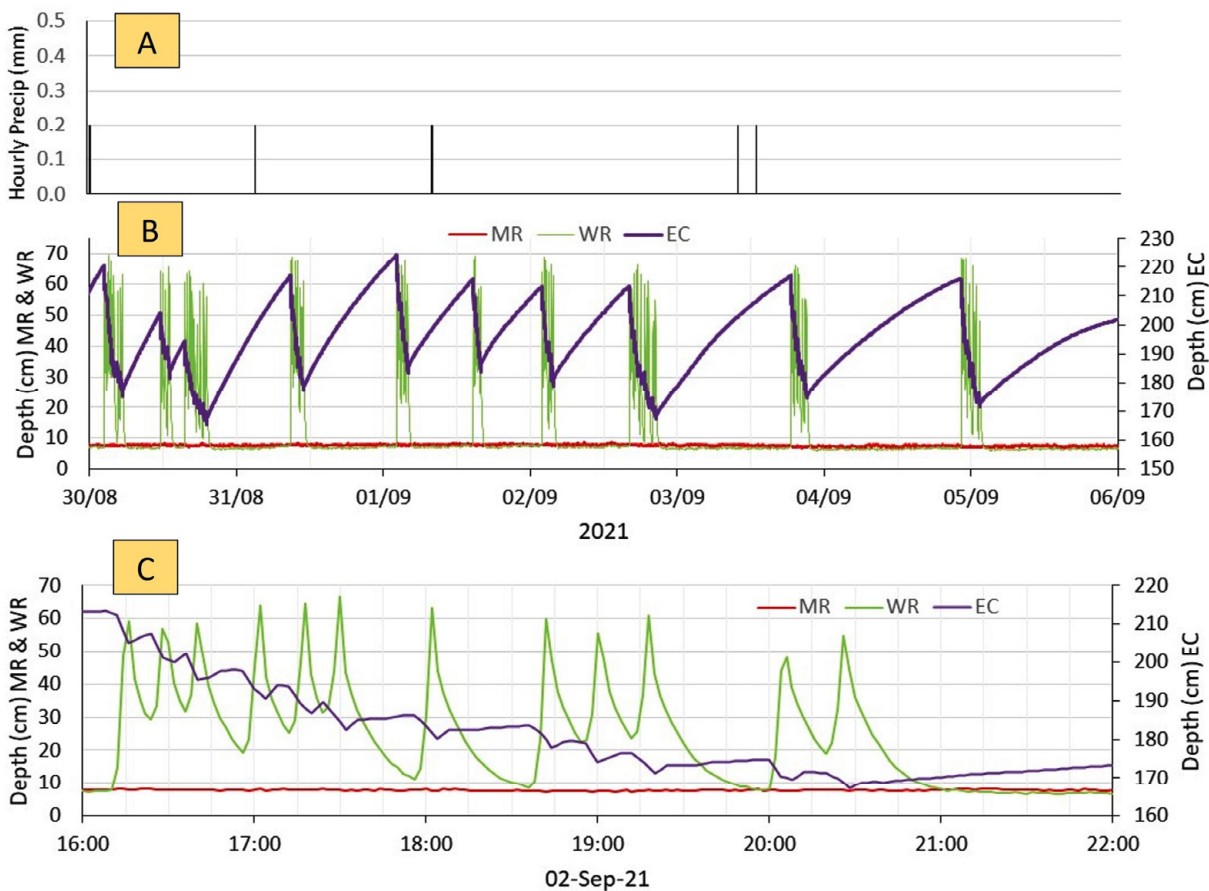

**Figure 12.** (**A**) hourly precipitation and (**B**) two-minute water depths at East Canal, Main Rising and Whirlpool Rising, from 30 August to 6 September 2021. (**C**) Detail for the period 16:00–22:00 on the 2 September 2021 during which there was no precipitation.

**Table 2.** Peak–trough cycles at EC, 30 August–5 September 2021 (these are the 10 events at EC shown in Figure 12).

| Event | Maximum | | Minimum | | Freq. | | Fall-Time | Fall | Rate | Rise-Time | Rise | Rate |
|---|---|---|---|---|---|---|---|---|---|---|---|---|
| | Time | cm | Time | cm | Peak | Trough | hh:mm | (cm) | cm/min | hh:mm | (cm) | cm/min |
| 1 | 30/08/21 02:20 | 220.6 | 30/08/21 05:20 | 175.3 | | | 03:00 | 45.3 | 0.25 | | | |
| 2 | 30/08/21 11:32 | 204.1 | 30/08/21 13:04 | 181.3 | 9:12 | 7:44 | 01:32 | 22.7 | 0.25 | 6:12 | 28.8 | 0.08 |
| 3 | 30/08/21 15:30 | 194.4 | 30/08/21 19:04 | 165.3 | 3:58 | 6:00 | 03:34 | 29.1 | 0.14 | 2:26 | 13.0 | 0.09 |
| 4 | 31/08/21 08:52 | 217.2 | 31/08/21 09:24 | 198.6 | 17:22 | 14:20 | 00:32 | 18.6 | 0.58 | 13:48 | 51.9 | 0.06 |
| 5 | 01/09/21 02:08 | 224.3 | 01/09/21 04:00 | 183.4 | 17:16 | 18:36 | 01:52 | 40.9 | 0.36 | 16:44 | 25.7 | 0.03 |
| 6 | 01/09/21 14:40 | 215.9 | 01/09/21 15:52 | 183.9 | 12:32 | 11:52 | 01:12 | 32.0 | 0.44 | 10:40 | 32.5 | 0.05 |
| 7 | 02/09/21 01:52 | 213.2 | 02/09/21 03:44 | 178.7 | 11:12 | 11:52 | 01:52 | 34.6 | 0.31 | 10:00 | 29.3 | 0.05 |
| 8 | 02/09/21 16:08 | 213.4 | 02/09/21 20:28 | 167.3 | 14:16 | 16:44 | 04:20 | 46.1 | 0.18 | 12:24 | 34.7 | 0.05 |
| 9 | 03/09/21 18:32 | 216.9 | 03/09/21 21:08 | 174.6 | 26:24 | 24:40 | 02:36 | 42.3 | 0.27 | 22:04 | 49.6 | 0.04 |
| 10 | 04/09/21 22:20 | 216.0 | 05/09/21 01:24 | 171.5 | 27:48 | 28:16 | 03:04 | 44.5 | 0.24 | 25:12 | 41.4 | 0.03 |
| | maximum | **224.3** | | **198.6** | **27:48** | **28:16** | **4:20** | **46.1** | **0.58** | **25:12** | **51.9** | **0.09** |
| | minimum | **194.4** | | **165.30** | **3:58** | **6:00** | **0:32** | **18.6** | **0.14** | **2:26** | **13.0** | **0.03** |

### 3.2.2. Subset 2: 27 September–3 October

Hourly rainfall from 27 September to 3 October are plotted in Figure 13A, and 2-min resolution water depths at EC, MR and WR are in Figure 13B. Over the 2 weeks prior to 27 September, there was only 9.8 mm of rain, of which 3.2 mm fell in the second week. During this period, water depths at EC fell slowly with no rhythmic changes, while depths at both MR and WR were virtually constant. Between 05:00 and 10:00 on 27 September, an intense storm delivered 24 mm of rain, of which 13.8 mm fell between 06:00 and 07:00. Between 07:20 and 08:40, water depths at EC rose linearly from 160 to 192 cm, and a peak–trough cycle was initiated. Between 08:40 and 08:46, the depth at WR increased from 4.9 cm to 61.1 cm, and a peak–trough cycle was initiated that was the reverse of the cycle at EC (i.e., each peak at EC corresponded with a trough at WR). This continued until 20:14, when rhythmic changes ceased at both EC and WR. From 20:14 until 02:48 on 28 September, water depths at EC increased slowly, whilst at WR depths remained at base level. After 02:48, a further interval of rhythmic depth change was initiated at both EC and WR, which continued until 09:30.

This was followed by another interval during which water depths at EC increased slowly, whilst those at WR remained at base level. Rhythmic pulsing began again at 14:02, and between 18:52 and 19:14 there was a sharp rise at EC which was followed by a slow recession interrupted by two small but sharp rises at 01:52 (8 min) and at 07:40 (10 min) on 29 September. Rhythmic changes were superimposed onto the recession and followed the same pattern as previously, each peak at EC coinciding with a trough at WR. Between 16:40 and 23:40 on 30 September, water depth at EC increased by 242.6 cm (0.58 cm/min) to a peak of 450.5 cm and then declined to a trough of 412 cm at 04:14 on 1 October before increasing again to a peak of 633.8 cm at 12:28 on 1 October. The depth changes on 1 October are shown in more detail in Figure 13C and indicate that as the depth increased at EC the amplitude of the rhythmic cycles at WR decreased and that when the depth at EC exceeded 600 cm the depth changes at WR are markedly reduced and may essentially be instrument noise. After the EC peak, the depth fell steadily for almost 24 h, and when it had passed 585 cm the peak–trough cycle at WR began again, the amplitude increasing as the depth at EC fell. At 12:04 on 2 October, there was a trough at EC, and as the depth began to steadily increase there was a repeat of the amplitude reduction in the WR peak–trough cycles, which were largely less than +/− 5 cm at EC depths above 590 cm.

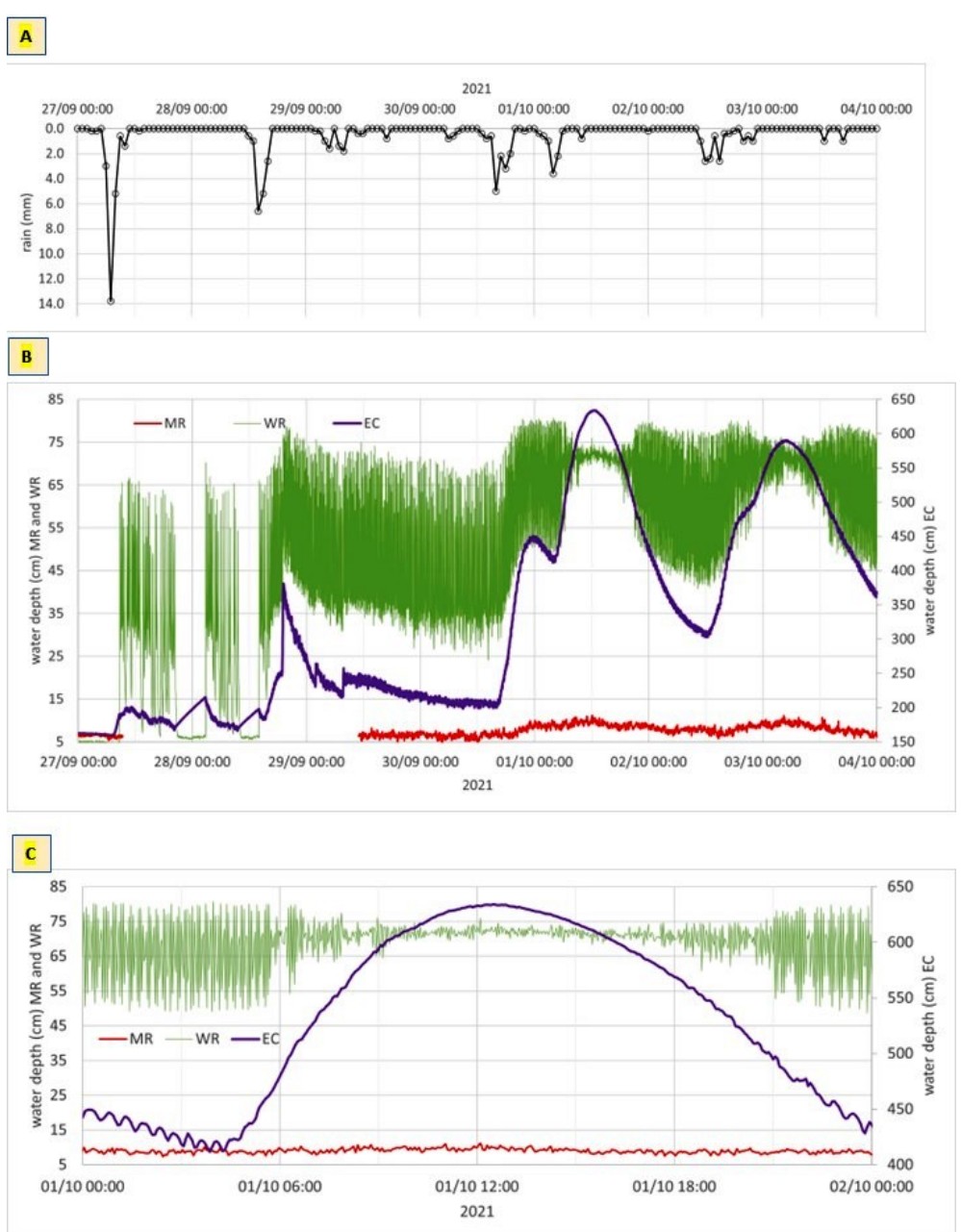

**Figure 13.** (**A**) Hourly precipitation (circles are the actual data points) and (**B**) water depths at East Canal (EC), Main Rising (MR) and Whirlpool Rising (WR) from 27 September to 3 October 2021. (**C**) shows more detail for the 1 October.

## 4. Discussion

The high-resolution in situ data presented above reveal a complex suite of hydraulic and hydrological responses to water inflow in the Castleton karst. These responses are a function of: (1) upstream water delivery and (2) the internal geometry and configuration of the conduit network. The first reflects the characteristics of individual rain events (total amount and intensity), antecedent conditions (state of soil moisture stores and underground reservoirs) and recharge pathways (allogenic and autogenic). In total, Figures 6–13 suggest a level of complexity in the Castleton karst that (to the authors' knowledge) seems greater than in any other published study.

While rhythmic karst springs have been described in the literature [3–6,8–11] and modelled [7,8] our results reveal three categories of hydrological behaviour: (1) a sustained linear or curvilinear increase or decrease in water depth; (2) a nonlinear rhythmic (wave-

like) pattern of varying water depth with sequential peaks and troughs; and (3) an episodic and rapid change in water depth which rises to a plateau, or decreases to an elongated trough, at the end of which there is a further rapid decrease or increase. The first type of behaviour is common in surface and underground streams and is driven by recharge. Systems with concentrated recharge are likely to produce more spikey hydrographs than those where recharge occurs over a more dispersed area. The second type of behaviour is less common but has been explained by invoking the operation of a siphon (e.g., [1,5,6]), of which two sub-types are evident at Castleton: regular and intermittent. The third type of behaviour is not thought to have been described previously other than in the Castleton karst [12]. In the following sections, we explore these mechanisms in detail.

*4.1. Rhythmic Depth Changes and Siphons*

At Castleton, periods of linear depth change (not described in detail here) are separated by wave-like oscillations that in some cases form part of a continuous series of peaks and troughs (Figure 7B, Figure 8 (after 12:00 27 February) and Figure 9) or an intermittent series in which peaks and troughs are separated by periods when water depths remain at base level (e.g., Figures 11B and 12). Two characteristics of the continuous series are: (a) peak and trough amplitude and frequency vary over time; and (b) the amplitude and frequency at MR differ from WR, although dye tracing indicates a physical connection between the two risings [15,16]. This level of complexity has not previously been described and requires further investigation (e.g., examining water temperature). However, it is most probably due to the effect of siphon(s) (see Appendix A) as flow through parts of the conduit network changes from open-channel flow (water movement under gravity) to a closed channel, with air expelled and water movement determined by pressure changes across the system (described by Bernoulli's Law).

The change in water flow from a gravity- to a pressure-dominated system may be a simple switch (i.e., open channel to closed conduit with increasing water inflow that reverts to open channel as inflows subside). However, during some events there may be multiple switches between flow mechanisms as water passes through the system. The situation is further complicated by the structural complexity of the karst system, with a network of vadose and phreatic conduits that vary in their connectivity as local water levels rise and fall [16]. Hence, there are likely to be multiple siphons, that are connected both 'in parallel' and 'in series' and which together represent the wider three-dimensional architecture of the karst network.

The potential circumstances in which siphon actions may develop can be illustrated by considering water movement through the conduits shown in Figure 4. While a siphon effect can be found in the simple situation outlined in the Appendix A (where a water-filled conduit connects a water reservoir to an outlet), it can also develop along individual conduits, when air is expelled due to water inflow. One example is the double ∩ shape in the conduit immediately upstream of WR (indicated by the red rectangle in Figure 4). Here, the tops of the bends are connected by a small passage, and if the latter fills with water, this 'unit' may behave as a single siphon, while during open-channel conditions, flow through the double bend (under gravity) may produce pulsing of flow through the conduit. Given that a significant part of the active conduit system at Castleton has yet to be explored, and a siphon effect can operate over many different scales, it is not possible at present to explain all the rhythmic behaviour described in Section 3. Moreover, a number of other processes add further complexity.

The hydrographs when intermittent rhythmic flow occurs are similar to the Type (a) hydrograph described in [1] and modelled by [2,7] but they are more complex in that the hydrographs in the literature all show depth increasing to a single peak before falling back to base level, whereas at WR (the only site with this behaviour) the base-level periods are separated by a series of peak–trough cycles. Studies [1,2,7] all attribute the rhythmic behaviour to the operation of siphons, but the pattern at WR (Figures 11B and 12) also resembles the reciprocating behaviour first described by Atwell in 1732 [6]. At Castleton,

there is also a clear link between the intermittent behaviour at WR and water depth at EC (Figure 12). Water depth at EC rises from base level to a peak and then drops rapidly. As water depths rise at EC, WR is quiescent, but the drop triggers a peak–trough cycle that ends as soon as water depths at EC start to increase again. While this may reflect the operation of a siphon or siphons, there is no consistency in amplitude or frequency. An alternative (or complementary) explanation is that sediment at the base of some of the phreatic loops may reduce the hydraulic conductivity. As water depths at EC increase, the pressure rises to a point when the sediment is displaced, triggering a series of peak–trough cycles. As the pressure at EC subsequently decreases, the velocity through the conduit decreases and sediment begins to settle, reducing the hydraulic conductivity to a point where flow is restricted and water depths begin to increase again at EC.

*4.2. Episodic Depth Changes and Sediment*

As suggested in Section 3, the episodic events are distinguished by: (a) a sustained period of time when water depth is broadly constant at a depth below the ambient base level (an elongated trough) or, less commonly, (b) a sustained period of time when water depths are maintained at a high elevation without any peak–trough cycles (a plateau). During the 2012–2015 data collection period, three types of episodic event are evident, all at MR: (1) two events in which a plateau with depth > 220 cm at MR is followed by a trough with rhythmic oscillations during recovery (e.g., Figure 7); (2) three events where the sequence is (a) broadly constant water depth at MR (in the range 45–53 cm), (b) a trough, and (c) a recovery peak (e.g., Figure 8); and (3) broadly constant water depth < 30 cm at MR interrupted by repeated troughs (e.g., Figure 9c).

The type 1 events start with a rapid (c. 1 cm/min) increase in depth to a plateau at c. 220 cm, after which the depth increases more slowly to a peak. After the peak, water depths initially fall slowly before the plateau ends with a rapid (c. 5 cm/min) fall to c. 20 cm. This depth is maintained for ~2 h before an even more rapid rise (c. 8 cm/min) after which there are 1–4 peak–trough cycles with large amplitude (c. 100 cm) and a frequency of 30–60 min. The type 2 events have no pre-event peak, and instead water depths fall rapidly (0.6–2.0 cm/min) from a base level at 45–51 cm down to c. 20 cm and remain at this depth for 3 h 50 m to 8 h 10 m. There is then a rapid rise (0.50–1.15 cm/min) to a peak of 89–99 cm, much higher than pre-event, after which there is a slow recession to the same depth as pre-event. In some cases, rhythmic peak–trough cycles are superimposed on the recession (e.g., Figure 8). Type 3 events, which all occur at MR depths < 30 cm, are similar to the type 2 events (in that there is no pre-event peak before the depth falls), but they differ in that at the end of the elongated trough water depths increase to a level that is only 1–3 cm above the pre-event level (e.g., Figure 9c). Each of the three event types at MR is accompanied by a different response at WR. During type 1 events, rhythmic pulsing at WR coincides with the MR trough period; during type 2 events, the MR trough is accompanied by a very slow (max 3 cm) increase in depths at WR with a peak that coincides with the end of the first post-trough depth increase at MR; and in type 3 events there is no change in depth at WR. The contrast between the three event types is also seen (in a more subdued form) in the depth hydrographs at RW, SM and at PW, the Peakshole Water weir situated downstream of PCR, SM and RW (Figure 3).

In 1987, three 'anomalous discharge events' were identified at PW [12] using data extracted manually from stilling well charts. While the data resolution was relatively coarse, two of the hydrograph shapes ([12]; Figure 2A,B) are almost identical to those of the 2012–2015 MR type 2 episodic events described above. It was estimated [12] that during the 1987 events 2700 m$^3$ of water were 'lost' during the trough but only 1500 m$^3$ gained in the subsequent peak. The 2-min resolution data collected for this paper allow a similar assessment of one type 2 episode (Figure 14). The 2-min water depth measurements at PW were converted into discharge using the weir rating equation and a line was projected forwards at the average pre-event discharge (471 L/s; 'base' in Figure 14). The volume below the line during the trough is estimated to be 3605 m$^3$, and the volume above the

line during the subsequent peak is 3621 m$^3$. Given the potential errors, particularly those due to rhythmic pulsing on the PW recession curve, these estimates are remarkably close and suggest that a volume of water was dammed up and then released. Before discussing possible mechanisms, it should be noted that the trough at PW has a more complex form than the MR trough, most likely due to downstream flow routing along the Speedwell Cavern vadose conduit.

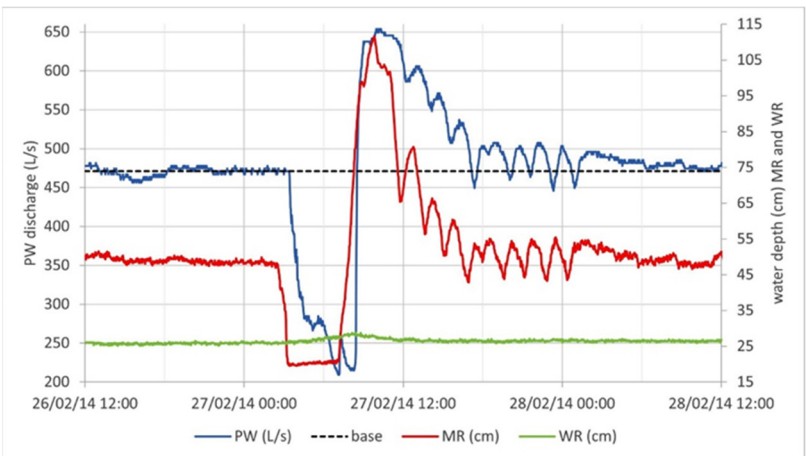

**Figure 14.** Water depth at MR and WR and discharge of PW, 12:00 26 February to 12:00 28 February 2014 (for precipitation, see Figure 8).

Bottrell and Gunn [12] suggested that the hydrographs in their Figure 2A,B reflected flow switching between MR and WR due to sediment movement in the phreatic part of the system, and they produced a simple model to predict the system response to subsequent flow-switching events. No in-cave data were available at the time, although the authors noted that both MR and WR exhibited "ebbing and flowing". Our in-cave data for water depths at MR and WR show that flow switching did not take place during the three observed type 2 episodes, as there was only a small (<3 cm) increase in depth at WR. However, there is evidence for flow switching during the two type 1 episodes, as in both cases the fall in depth at MR, that marked the start of the elongated trough, was accompanied by a rise in depth at WR and the start of a rhythmic peak–trough cycle. As the elevation of the water surface at WR is ~11.5 m above the base elevation at MR (Figure 4), the following hypothesis is advanced. During type 2 and type 3 episodes, sediment accumulates at the base of one or more of the phreatic loops in the conduit that exits at MR. This reduces hydraulic conductivity along the conduit and reduces flow at MR and results in an increase in the head upstream of the sediment blockage. In the type 2 events, this increase is sufficient to produce the small increase in depth seen at WR: i.e., the increase in head is sufficient to force water through the blockage, restoring the hydraulic conductivity of the conduit. In the type 1 events, a similar mechanism operates, but in this case the increase in head is sufficient for flow and depth at WR to increase.

A similar mechanism has been proposed to explain episodic discharge from Big Spring in Kings Canyon National Park, CA, USA [9–11]. Water depth observations in the Z Room, Lilburn Cave and discharge measurements at the spring "suggest a single conduit containing a sediment plug in the lowest sump that stochastically blocks the flow path creating ebb and flow discharge cycles. A larger cross-sectional area is present above the sump that retains most of the sediment because of a lower velocity" [10] (p. iv). The latter observation is relevant to MR, as at its furthest explored point, 74 m below the rising, water has been described as 'boiling-up' through a floor of liquid sand from a slot around 2 m wide [18], and from this point the water rises vertically 40 m up the New Leviathan shaft (Figure 4). Observations following flood events of fresh sand deposited in the Speedwell Cavern vadose streamway downstream of MR demonstrate that in these events some sediment is mobilised and evacuated from the sump. However, during smaller events the

velocity may be sufficient to mobilise sand-sized sediment, but the flow is insufficient to carry it to the top of the shaft(s), and the sediment will settle as the velocity falls. Any sediment lost from the phreatic conduits will be replaced by inputs from the allogenic streams [23].

A more substantial increase in the hydraulic conductivity of the sediment plug at the base of New Leviathan, or possibly at the base of a similar upstream phreatic loop, is also the most likely explanation for the switch from MR being the dominant input to Speedwell Cavern (2012 to 2015 data and observations by cavers up to 2020) to WR being the dominant input from 2021 (initial report by cavers followed by data in this paper). In January 2021, 182 mm of rain fell in the Castleton catchment over 10 days during Storm Christoph and subsequent un-named storm events, and it is hypothesised that this mobilised sediment both in the allogenic catchment and underground. As flows decreased after the storm, a greater thickness of sediment than had previously been the case accumulated at the bottom of phreatic loops, which resulted in the substantial decrease in flow from MR and corresponding increase in flow from WR. This could represent a permanent change to the system hydrology, as on 2 Nov 2021 the water elevation in EC reached 264 m asl, a head of 32 m above MR (Figure 4) without displacing the sediment plug. However, given past observations of flow switching, it is likely that in future the hydraulic conductivity of the plug will increase again, or it will be completely breached, whereupon MR will again be the dominant input.

## 5. Conclusions

High-resolution (1 min–4 min) in situ monitoring of water depth at three underground sites has revealed a range of rhythmic and episodic responses to water inflow. These are a consequence of the complexity of the conduit networks in the karst, both horizontally (as demonstrated by water tracing experiments that confirm both convergent and divergent flow [19,20]) and vertically (as shown by cave surveys such as Figure 4). Changes in connectivity between individual conduits add to the complexity. These changes in connections occur when lower elevation conduits are surcharged or when their hydraulic conductivity is reduced due to sediment accumulation, forcing water through higher elevation conduits. Short-term variations in water depths appear to be more complex than documented elsewhere in the literature to date.

Depth hydrographs from EC at the upstream end of a phreatic conduit, and from MR and WR at the downstream ends of phreatic conduits, reveal characteristic responses to water inflow. Flow from MR and WR is then routed down a vadose conduit (free surface flow) and enters another phreatic conduit (pressure flow) before emerging at the RW and SM springs. In 2022, a depth logger was installed upstream of the final phreatic conduit to obtain additional data on flow routing. This will form part of our further analysis, quantification, and modelling of the Castleton data set introduced in this paper. However, the initial results highlight the potential for in-cave monitoring to aid understanding of spring hydrographs.

**Author Contributions:** Research conceptualization; methodology and fieldwork: J.G.; C.B. contributed to the interpretation of results and assisted with the preparation of the manuscript. All authors have read and agreed to the published version of the manuscript.

**Funding:** The British Cave Research Association Cave Science and Technology Research Fund provided grants to Nigel Ball and to John Gunn for the purchase of some of the equipment used in this project. No other external funding was received.

**Data Availability Statement:** The supporting data can be obtained from the authors on request.

**Acknowledgments:** The research was made possible by the support of many members of the caving community and in particular the late Nigel Ball who initiated the project and undertook most of the underground logger downloads between 2012 and 2015, assisted mainly by David Shearsmith and Nick Coward. David Shearsmith and Mark McAuley have assisted JG with underground data collection since June 2021. Grateful thanks are extended to John Harrison for permission to enter Peak

and Speedwell Caverns and to place instruments at PCR and SM; to Vicky Turner for permission to access RW, to David Hubble for permission to build the weir and install instruments at PW, and to Jonathan Down for permission to locate a rain gauge at Coalpithole Mine. Figure 1 was drafted by Andrew Farrant and Figure 2 by Ellen Lynch. Rob Eavis is thanked for compiling the surveys that form Figure 4, and the late Nigel Ball provided the photographs used in Figure 5.

**Conflicts of Interest:** The authors declare no conflict of interest.

### Appendix A

The operation of a siphon in karst can be illustrated using Figure A1 (Mather's Figure 6: [5]) and considering flow through the conduit connecting a water reservoir in the karst, to an outlet such as a spring, or larger conduit. If water levels in the reservoir rise from H2 to H1, there will initially be no flow through the conduit, as the passage initially rises on leaving the reservoir. If reservoir water levels continue to increase above H1, then water will start to flow through the conduit to its outlet. As air is expelled from the siphon conduit (when water levels rise above H1), water will continue to flow through the conduit until reservoir water levels fall to H2, when air can re-enter the conduit. At this point, flow through the siphon conduit will cease until water levels in the reservoir return to H2. This may lead to pulse of water flow along the siphon conduit, if the rate of water inflow to the reservoir is maintained, with a pulse cycle that reflects the volume of water inflow, the configuration of the siphon conduit and reservoir, and the difference in height between H1 and H2. The duration of the pulsing cycle will depend upon the continued inflow of water to the water reservoir. At the conclusion of the event, the pulsing cycle will change, as the rate of water inflow to the reservoir falls, and water levels subside to H2.

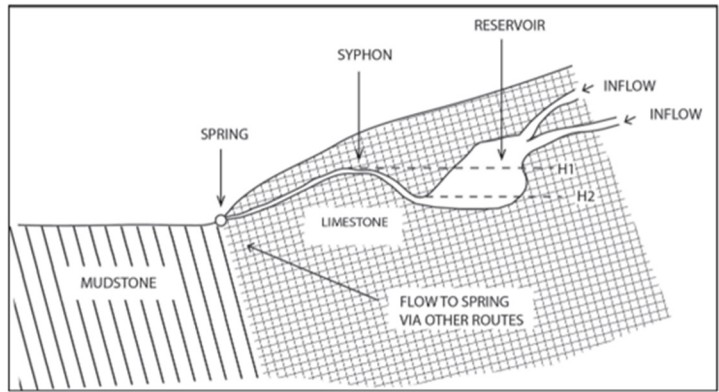

**Figure A1.** Schematic illustrating the situation in which a siphon might develop in karst [5].

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
