# Peer review of "Characterising Rhythmic and Episodic Pulsing Behaviour in the Castleton Karst, Derbyshire (UK), Using High Resolution in-Cave Monitoring"

_water, doi:10.3390/w15122301_

Round 1

Reviewer 1 Report

Dear Authors,

I found this paper very interesting and I do not have any further suggestions, apart from providing graphs and maps that have better resolution.

Author Response

We thank the reviewers for their helpful feedback and make the following responses to their comments.

REVIEWER 1

I found this paper very interesting and I do not have any further suggestions, apart from providing graphs and maps that have better resolution. The authors are pleased that the reviewer found the paper of interest. Where possible we have improved graph and map resolution but it should be noted that all graphs and maps are markedly improved if viewed at 200%.

REVIEWER 2

Line 15. Be more specific when you describe the flow dynamics techniques in the abstract. Provide detail and explanation.The authors are not clear what the Reviewer requires. Also, the Abstract is presently at the 200 word limit and to add detail and explanation would increase its length above that limit.

Line 94. Clearly state the specific objectives of your research using numbers (i), (ii), and (iii) at the end of your introduction. As requested we have added the Roman numerals.

Lines 97-170. Here or above you need to mention that the Carbonifeous Limestones are highly karstified in England. Please, add the following references below:

- Medici, G. and West, L.J., 2021. Groundwater flow velocities in karst aquifers; importance of spatial observation scale and hydraulic testing for contaminant transport prediction. Environmental Science and Pollution Research, 28(32), pp.43050-43063.

- Worthington, S.R. and Ford, D.C., 2009. Self‐organized permeability in carbonate aquifers. Groundwater, 47(3), pp.326-336.

The words “highly karstified” have been inserted on lime 102. However, the 2 references suggested by the Reviewer do not contain any information on the Peak District karst and instead 2 references that describe the Peak District karst hydrogeology have been included. A phrase, and supporting reference, has been added to make it clear that the Peak Limestone Group is part of the Carboniferous Limestone Supergroup.

Lines 97-170. Provide detail on the groundwater flow dynamics, principal direction and relation with the topography. The authors are not clear what the Reviewer is requesting regarding ‘groundwater flow dynamics’. Lines 125-127, and Figure 2, show the underground flow direction and this is totally unrelated to surface topography (other than the fact that sinks are higher elevation than springs) as can be seen from the contour lines on Figure 3. A sentence has been added to this effect (lines 127-129).

Lines 97-170. You provided detail on the stratigraphy, but not on the presence of faults. Any link between caves and faults? There is little by way of structural guidance of underground drainage in the Castleton karst. Figure 2 shows the principal mineral rakes which are fault aligned (a sentence has been added to the caption to this effect).

Lines 227-241. Words in bold, please fix the editorial issue. The words were emboldened to indicate they have specific definitions as per the following text.

Lines 521-715. The discussion is very long. To justify the choice, add more literature to support your argument. Alternatively, you can make the discussion shorter. We have re-written the discussion and shortened it by c. 10%). Unfortunately there is very little literature on rhythmic and episodic pulsing and this was one of the drivers for our paper.

Lines 718-721. Long sentence in a short conclusion. I suggest to split the sentence in two parts adding relevant detail. The authors agree that this is over-long and have used this opportunity to re-write the Conclusions.

Line 723. Better “large parts” than “majority”? Just a suggestion, I’m not a native speaker, just PhD and postdoc in England! In this context the authors argue that “majority” is the better term.

Line 774. Add more references as suggested above. Done

Figures and tables

Figure 10. Increase size, figures and labels

We have revised the font size (increasing by c. 10%)

Reviewer 3

The manuscript entitled “Characterising rhythmic and episodic pulsing behaviour in the Castleton karst, Derbyshire (UK), using high resolution in-cave monitoring”, by J. Gunn and C. Bradley, presents an interesting work. In general, the manuscript should be acceptable for publication but some problems must be repaired prior to publication. Some suggestions are as follows:

  1. The abstract should state briefly the purpose of the research, the principal results and major conclusions. An abstract is often presented separately from the article, so it must be able to stand alone. The authors consider that the present abstract fulfils these requirements
  2. It would be useful to be described the aim of this paper. The aims are described in the final sentence of the introduction.
  3. You could enrich the scientific literature. As we explain in the paper, there is very little literature relating to the topic covered in our paper and we have cited all relevant papers. We do not believe that citing papers simply to make a document look more impressive is a valid approach.
  4. Please justify convincingly why this manuscript (method, thematology etc) connected with WATER’s content and scope. Perhaps the using of proper literature from this journal would be helpful. Eg:

- Alexakis, D.E. Anthropogenic and Geo-Environmental Impacts on the Hydrosphere: Diagnosis, Monitoring, Assessment, and Sustainable Management. Water 202315, 1390. https://doi.org/10.3390/w15071390. The Reviewer appears to have failed to appreciate that this paper is submitted as part of a special issue on “Cave Waters: Modern Perspectives for Short to Long-Term Environmental Monitoring”. The paper by Alexakis does not appear to have any relevance to this topic but the authors have added a reference to a paper by Skoglund et al that has been published as part of the Special Issue and is relevant to the topic covered in our paper.

  1. Please use coordinates in all maps. As no coordinates are used in the paper they are not necessary on the maps but Figure 1 has coordinates.
  2. When you are using coordinates, please do not use “North Arrow”. This is a mistake in cartography. With respect we disagree

Reviewer 2 Report

General comments

Research on karst hydrology based on a robust dataset that fits the scope of the special issue. However, some detail is missing and the manuscript needs some re-organization. All the specific comments need to be addressed before publication

Specific comments

Line 15. Be more specific when you describe the flow dynamics techniques in the abstract. Provide detail and explanation

Line 94. Clearly state the specific objectives of your research using numbers (i), (ii), and (iii) at the end of your introduction

Lines 97-170. Here or above you need to mention that the Carbonifeous Limestones are highly karstified in England. Please, add the following references below:

- Medici, G. and West, L.J., 2021. Groundwater flow velocities in karst aquifers; importance of spatial observation scale and hydraulic testing for contaminant transport prediction. Environmental Science and Pollution Research28(32), pp.43050-43063.

- Worthington, S.R. and Ford, D.C., 2009. Self‐organized permeability in carbonate aquifers. Groundwater47(3), pp.326-336.

Lines 97-170. Provide detail on the groundwater flow dynamics, principal direction and relation with the topography

Lines 97-170. You provided detail on the stratigraphy, but not on the presence of faults. Any link between caves and faults?

Lines 227-241. Words in bold, please fix the editorial issue

Lines 521-715. The discussion is very long. To justify the choice, add more literature to support your argument. Alternatively, you can make the discussion shorter

Lines 718-721. Long sentence in a short conclusion. I suggest to split the sentence in two parts adding relevant detail

Line 723. Better “large parts” than “majority”? Just a suggestion, I’m not a native speaker, just PhD and postdoc in England!

Line 774. Add more references as suggested above

Figures and tables

Figure 10. Increase size, figures and labels

Author Response

(The authors gave the same response as above.)

Reviewer 3 Report

Summary

This is a data-rich, model-poor paper that summarizes a suite of depth and related hydrologic measurements in the Castleton karst, Derbyshire, UK.    The paper is valuable for its demonstration, at different time scales, that patterns of flow in the cave system are not always simply connected to precipitation events. These patterns suggest the operation of different reservoirs of different sizes that need to be filled to flow and that behave like capacitors when they are close to filled.  The reservoirs seem to communicate, but lag times and flow patterns are both rhythmic and variable in ways that seem to repeat over time.  All of the responses described in the paper are related to precipitation events and thus infiltration in the broad sense, but the system is complex! The authors invoke “siphon behavior” or the possibility of sediment plugging and unplugging to help explain changes in water depth that they observe, Both causes seem probable and down gradient observations will necessarily reflect changes in a variety of systems (different volumes and effective hydraulic conductivity) over time. The authors do not attempt to model their data, perhaps because they have no direct measurements or sites that reflect simple upgradient hydrology.

I summarize, in no particular order of importance, some of the major suggestions I have about the text and figures below and have made numerous comments on the .pdf of the manuscript.

1.    Are you certain that you cannot characterize different portions of the hydrograph time periods by characteristic spacing of peaks?   At the very least you can characterize different interval/amplitude patterns and their lags from the centroid of the precipitation events. Your dye tests tell you something about travel times and thus about lags in part of the system. You record some of these observations in Table 2, but do not attempt to group them.

2.    In some ways Figure 10 summarizes many of the patterns you discuss in the text. You need to add some form of the precipitation hydrograph for each of these time series.  It would also help to “show” some form of antecedent moisture or water level in a well finished in something other than carbonate.

3.    Results.  There is too much descriptive text in the results and there will be even more if some of the “discussion” moves into the results, as I suggest below.  The excellent figures and table (there could be one more) obviate the need for so many lines of text. Readers will note starting and stopping points and rhythmic patterns on their own.

4.    Are there anthropogenic influences (pumping; discharge) that might influence some of your observations.

5.    Discussion.  This is where the authors should begin to explain the extensive observations they have collected from what is surely a complex system.  However, too much of the “discussion” represents additional results, and should be moved there.

Reviewer 4 Report

The manuscript entitled Characterising rhythmic and episodic pulsing behaviour in the Castleton karst, Derbyshire (UK), using high resolution in-cave monitoring, by J. Gunn and C. Bradley, presents an interesting work.

In general, the manuscript should be acceptable for publication but some problems must be repaired prior to publication. Some suggestions are as follows:

  1. The abstract should state briefly the purpose of the research, the principal results and major conclusions. An abstract is often presented separately from the article, so it must be able to stand alone.
  2. It would be useful to be described the aim of this paper.
  3. You could enrich the scientific literature.
  4. Please justify convincingly why this manuscript (method, thematology etc) connected with WATER’s content and scope. Perhaps the using of proper literature from this journal would be helpful. Eg:

- Alexakis, D.E. Anthropogenic and Geo-Environmental Impacts on the Hydrosphere: Diagnosis, Monitoring, Assessment, and Sustainable Management. Water 202315, 1390. https://doi.org/10.3390/w15071390

  1. Please use coordinates in all maps.
  2. When you are using coordinates, please do not use “North Arrow”. This is a mistake in cartography.

Author Response

(The authors gave the same response as above.)
